# Review on Rail Damage Detection Technologies for High-Speed Trains

Yu Wang, Bingrong Miao *, Ying Zhang, Zhong Huang and Songyuan Xu

State Key Laboratory of Rail Transit Vehicle System, Southwest Jiaotong University, Chengdu 610031, China; yu2022wang@163.com (Y.W.)
* Correspondence: brmiao@home.swjtu.edu.cn

**Abstract**

From the point of view of the intelligent operation and maintenance of high-speed train tracks, this paper examines the research status of high-speed train rail damage detection technology in the field of high-speed train track operation and maintenance detection in recent years, summarizes the damage detection methods for high-speed trains, and compares and analyzes different detection technologies and application research results. The analysis results show that the detection methods for high-speed train rail damage mainly focus on the research and application of non-destructive testing technology and methods, as well as testing platform equipment. Detection platforms and equipment include a new type of vortex meter, integrated track recording vehicles, laser rangefinders, thermal sensors, laser vision systems, LiDAR, new ultrasonic detectors, rail detection vehicles, rail detection robots, laser on-board rail detection systems, track recorders, self-moving trolleys, etc. The main research and application methods include electromagnetic detection, optical detection, ultrasonic guided wave detection, acoustic emission detection, ray detection, vortex detection, and vibration detection. In recent years, the most widely studied and applied methods have been rail detection based on LiDAR detection, ultrasonic detection, eddy current detection, and optical detection. The most important optical detection method is machine vision detection. Ultrasonic detection can detect internal damage of the rail. LiDAR detection can detect dirt around the rail and the surface, but the cost of this kind of equipment is very high. And the application cost is also very high. In the future, for high-speed railway rail damage detection, the damage standards must be followed first. In terms of rail geometric parameters, the domestic standard (TB 10754-2018) requires a gauge deviation of $\pm 1$ mm, a track direction deviation of 0.3 mm/10 m, and a height deviation of 0.5 mm/10 m, and some indicators are stricter than European standard EN-13848. In terms of damage detection, domestic flaw detection vehicles have achieved millimeter-level accuracy in crack detection in rail heads, rail waists, and other parts, with a damage detection rate of over 85%. The accuracy of identifying track components by the drone detection system is 93.6%, and the identification rate of potential safety hazards is 81.8%. There is a certain gap with international standards, and standards such as EN 13848 have stricter requirements for testing cycles and data storage, especially in quantifying damage detection requirements, real-time damage data, and safety, which will be the key research and development contents and directions in the future.

**Keywords:** track rail; intelligent operation and maintenance; non-destructive testing; intelligence; life prediction

## 1. Introduction

With the rapid development of China's railway network, especially the proposal of "new infrastructure", the deployment of high-speed maglev trains with a speed of 600 km per hour has put forward higher requirements for the safety of railway tracks. As a new and important means of transportation, railways play an extremely important role. The safety and reliability of railways are receiving increasing attention from relevant research scholars, and the advancement of technology has led to higher requirements for the transportation functions of railways, especially for heavy-duty, high-speed, and other functions. Due to the contact friction between train wheels and rails, during heavy-load and high-speed movements, there is contact sliding friction between the wheels and rails. During the sliding friction process, various types of damage are easily generated on the surface of the rails. As long as there is damage on the surface of the rails, it is inevitable for the wheels to wear out. Some defects in the rails themselves can easily cause train derailment, resulting in unnecessary losses and threatening personal safety.

Regarding the detection of rail damage, experts and scholars at home and abroad have conducted in-depth research and proposed some detection methods and theories. However, with the increase in train speed, the requirements for maintenance and operation technology are also increasing. Currently, there are still some shortcomings in the rail damage detection methods, including hardware detection in vehicles and supporting infrastructure. For example, the current manual detection method has disadvantages such as a high false-detection rate, low efficiency, and significant subjective influence. The eddy current testing method has high requirements for its own materials, which must be conductive materials, and has disadvantages such as low detection speed, low efficiency, and susceptibility to magnetic fields. The X-ray detection method has high detection costs, low detection speed, and low efficiency, and radiation has a significant impact on human health. The ultrasonic detection method has low detection accuracy and cannot accurately detect the precise location of rail damage, and the detection speed is easily affected. It has a series of problems and disadvantages such as low detection accuracy for rail structures with complex structures. The detection speed of the leakage magnetic detection method is highly susceptible to the influence of magnetic flux, and there are certain requirements for the lift-off value of the rail surface, which also has a significant impact on the detection speed. Although machine vision inspection can quickly and efficiently detect surface defects on railway tracks in a non-contact manner, there are still certain limitations in detecting internal damage in railway tracks.

The comparison of common methods for detecting rail damage is shown in Figure 1. From Figure 1, it can be found that the common drawback of several detection methods is low detection efficiency. To solve the problem of the efficient detection of rail damage, it is necessary to collect and analyze real-time data during the operation of the rail, that is, monitor and provide real-time feedback on the status of the rail during operation, achieving zero delay in operation and maintenance.

As shown in Figure 2, the national railway passenger and freight volumes in the past 5 years have been increasing year by year, despite being affected by certain factors. This reflects the increasing importance of railway transportation, as well as the importance of the maintenance and operation of railway transportation. In the face of the current operation and maintenance of high-speed and ultra-high-speed trains, more efficient, intelligent, and digital non-destructive testing technologies and methods are needed to monitor the real-time status of rails and perform intelligent operation and maintenance.

How to quickly, accurately, and intelligently detect railway damage has always been an important challenge for many scientific researchers, and it is also a very important task in the railway maintenance process. Currently, there is a great demand for very complete

and intelligent railway maintenance, upkeep, and detection systems and equipment, the development of which is a difficult problem for many scientific and technical personnel to tackle. At present, the main method of rail inspection still relies on manual inspection, such as manual patrol, the scattering detection magnetic powder method, and other manual methods for inspection. Methods based on new eddy current meters, comprehensive track recording vehicles, laser rangefinders, thermal sensors, laser vision systems, laser radar, new ultrasonic detectors, rail inspection vehicles, rail inspection robots, laser vehicle-mounted rail inspection systems, track recorders, self-propelled carts, etc., have certain applications in rail inspection, but there are still certain limitations. Developing more efficient, reliable, fast, accurate, and intelligent systems and detection equipment is particularly important for ensuring the safety of railway track quality.

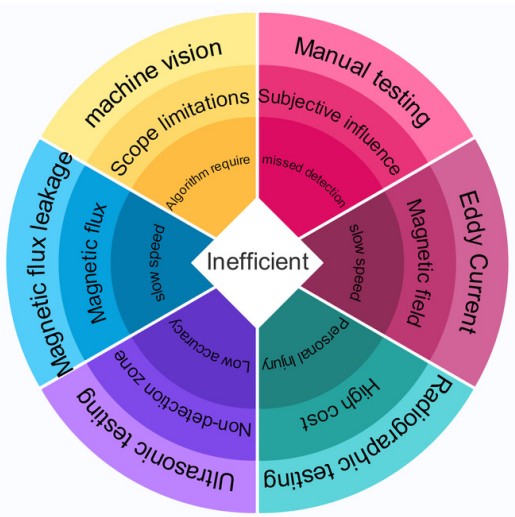

**Figure 1.** Comparison of shortcomings of common rail damage detection methods.

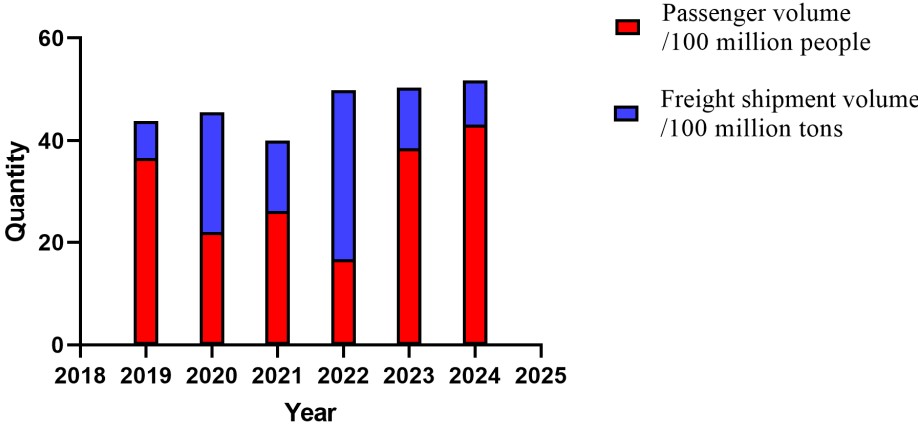

**Figure 2.** Passenger and freight volume of railway national railways in the past 5 years.

## 2. Research Status of Non-Destructive Testing for Rail Damage

As railways are an important means of transportation, track life detection and reliability analysis have gone from manual inspection to a series of detection methods and means such as radar technology detection, eddy current detection, machine vision, ultrasonic detection, etc. For common defects, cracks, and fasteners on the surface of railway tracks, detection is still quite time-consuming.

In recent years, with the rapid development of computer technology, image detection methods based on computer vision technology and deep learning have been used for railway inspection. For example, Reed et al. [1] proposed a method for region de-

tection with CNN features (R-CNN), which has significantly improved the accuracy of object detection and is considered a major breakthrough in the field of object detection. Neubeck et al. [2] constructed a Feature Pyramid Network (FPN), which is a universal feature pyramid extraction method located in deep convolutional networks, to help detection models detect feature objects on a large scale.

In terms of railway maintenance and upkeep equipment, developed Western countries often use handheld systems (such as ultrasonic speed measuring rods) or high-speed dual-use railway/road vehicles equipped with various non-destructive testing sensors for manual inspection. However, the railway standards and maintenance standards of different countries are different, which also leads to some difficulties and challenges in the inspection process of railway maintenance and upkeep equipment. Below, Figure 3 shows the detection standards for the direct relationship between railway track operating speed and tonnage in the UK.

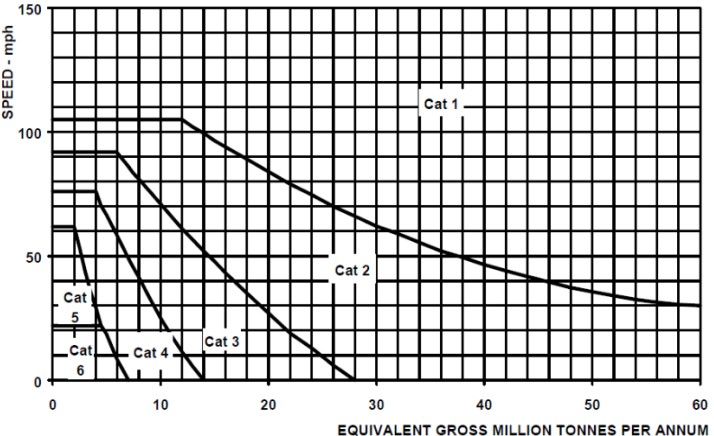

**Figure 3.** Test standard for the direct relationship between railway track speed and tonnage in the United Kingdom.

The train operation management automation system (COMTARC) of Japan's Shinkansen railway mainly uses fiber optic sensors to monitor foreign object intrusion limits. Each monitoring point determines whether intrusion limits have occurred and the degree of harm through the attenuation of optical signals. When the alarm control conditions are reached, there is no need for dispatch personnel to manually confirm, and the train speed limit control is immediately activated [3]. The continuous real-time tracking automation system (ASTREE) and train automatic control system (TVM-430) [4] of French railways mainly achieve the function of foreign object intrusion limit detection by setting up a protective monitoring alarm network. When the alarm control conditions are reached, the train speed limit is achieved, simultaneously using radar technology to detect whether passengers have intruded into the track space on the platform and issue an alarm. With the development of artificial intelligence technology, the combination of artificial intelligence and robotic technology can minimize the errors caused by manual operations in railway inspections. As shown in Figure 4, the ACFM [5,6] cane manual rail detector is used.

The ACFM probe inevitably causes lifting changes or magnetic changes along the inspection direction (due to residual stress) at higher speeds, which may lead to an increase in background signals. If the threshold is set too low, it may result in certain defects not being detected. On the other hand, setting the threshold too high depending on the SNR may result in multiple false alarms. Preliminary investigations into low-speed and high-speed ACFM measurements on steel rails indicate that it is difficult to adjust the threshold to detect all defects.

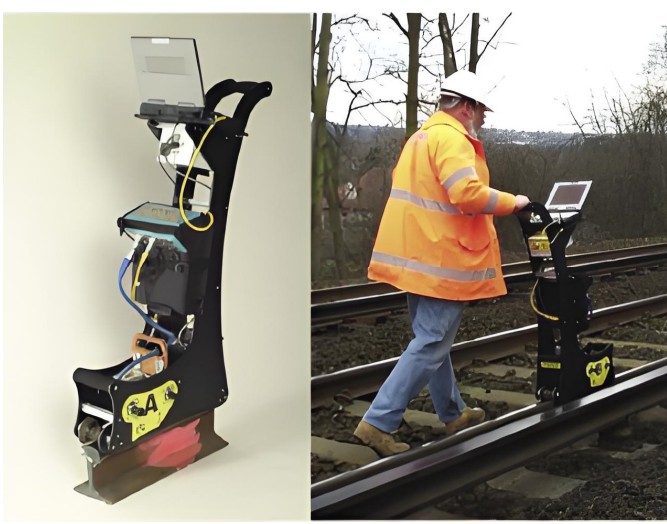

**Figure 4.** ACFM Hand Stick Manual Rail Tester.

In recent years, China's railway inspection technology has also developed rapidly. Hu Qingwu proposed a mobile binocular vision measurement model for railway building clearance which calibrates and constructs the spatial relative relationship of various elements in the stereo image through self-checking and multi-sensor system calibration. A building clearance calculation model based on the automatic fitting calculation of the spatial position of the centerline of the line was presented [7].

The traditional eddy current testing method mainly uses periodic pulse current signals as excitation sources to perform post-inspection or the online testing of materials (Figure 5). In 2006, Ding Tianhuai et al. used a time-division multiplexing-based eddy current array testing method to achieve the real-time monitoring of the position of large-area metal curved surface components using a flat flexible eddy current sensor array [8]. In 2017, Shi Tongyu et al. used an analytical model of a disc eddy current drive based on the equivalent magnetic circuit method [9]. In 2017, Cao Aisong studied the problem of using magnetic conductor ring excitation pulse eddy current detection technology to detect circumferential crack defects in small-diameter pipes [10].

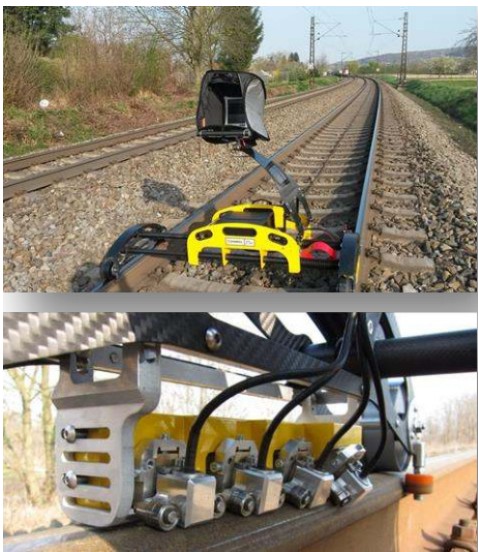

**Figure 5.** Manual eddy current detector.

The main principle of ultrasonic testing is to use the piezoelectric effect of piezoelectric chips to generate ultrasonic waves for the internal inspection of the specimen. In 2014,

Sohn H. and Zhao, Y. conducted qualitative testing on metal fatigue cracks using nonlinear ultrasonic modulation technology [11–13]. In 2017, Liu, Y. used Lamb waves to detect surface microcracks based on the characteristic that the influence of randomly distributed microcracks on the fundamental phase velocity of low-frequency SO can be ignored [14,15]. In 2017, Park, B. et al. proposed using multi-sensor ultrasonic pulse echo technology to study the flow characteristics of wind-driven surface water film [16,17]. In 2017, Ohara, Y. utilized ultrasound tomography technology to image the interior of materials [18–20].

The domestically produced GTC-80-II-J rail flaw detection vehicle has outstanding performance, with a maximum operating speed of 120 km per hour and a maximum continuous flaw detection speed of 80 km per hour. The GTC-80-II-J rail inspection car was developed by Baoji CRRC Times Engineering Machinery Co., Ltd., Shaanxi, China and officially put into use in 2023, breaking the technological monopoly of foreign countries in the field of high-end rail inspection equipment. In terms of rail geometric parameters, the domestic standard (TB 10754-2018) [21] requires a gauge deviation of $\pm 1$ mm, a track direction deviation of 0.3 mm/10 m, and a height deviation of 0.5 mm/10 m, and some indicators are stricter than European standard EN-13848 [22]. The vehicle conducts non-destructive testing by emitting and receiving ultrasonic waves through inspection wheels. A total of 18 probe chips and 26 ultrasonic channels are installed in the six inspection wheels under the vehicle, which can accurately detect steel rails in all directions with a damage detection rate of over 85%. To address the problem of detection deviation caused by rail wear and vehicle "snake like" motion, 3D laser track profile measurement technology is adopted, combined with adaptive wear compensation control, to dynamically adjust the position of the probe wheel, and the error can be controlled at the millimeter level. Its detection data support the synchronous storage and playback of A/B display signals, coupled with the automatic analysis of high-definition images of the rail surface, achieving the dual verification of "waveform+image" and greatly reducing the risk of missed detections. The diverse service environment of high-speed trains, with large speed fluctuations and unclear shock loads, leads to signals with strong nonlinearity [23–25].

Domestically, unmanned aerial vehicles equipped with laser radar, AI image recognition, and other technologies are used in track detection. A rail surface defect detection method based on unmanned aerial vehicle images has a recall rate of 93.75% and an accuracy rate of 93.6%. By combining 3DGIS and digital twin technology, air vehicle–ground collaborative inspection can be achieved to improve detection efficiency. However, in complex environments such as strong- or low-light conditions, defect features are easily masked during drone detection.

The training data for foreign AI models are larger, and the algorithms are more mature. For example, JR Kyushu in Japan uses AMD Kria K26 system modules combined with FPGA technology to achieve high-speed image processing, with an accuracy rate of over 85% in rail sleeper damage classification. AMD Kria K26 system module was developed and launched by AMD (acquired Xilinx, San Jose, CA, USA) and released in 2021. It focuses on edge computing applications and aims to provide flexible and high-performance solutions for industrial Internet of Things, machine vision, medical imaging and other fields. Multi-sensor fusion technologies such as laser radar, infrared, and ultrasound are commonly used abroad. For example, the French TGV detection system integrates subsystems such as track geometry measurement and contact network detection to achieve the comprehensive monitoring of railway facilities. However, there are shortcomings in the depth of multi-sensor data fusion in China, and there is a lack of a unified standard framework.

Electromagnetic testing is based on the principle of electromagnetic induction, mainly including magnetic particle testing, magnetic leakage testing, alternating current electromagnetic field technology (ACFM), and metal magnetic memory testing technology. In

2014, Wen Xiaode et al. [26] used magnetic flux leakage testing technology to detect surface crack defects. In 2017, He Yunze used the blind source separation method to process electromagnetic detection signals using BSS, which showed significant improvement compared with traditional methods [27].

Research has found that current railway track characterization defects such as rolling contact fatigue (RCF) cracks can be detected to a certain extent, but there are certain errors. For example, the ACFM cane manual rail inspection instrument has significant errors in detecting rail characteristic crack defects, with an automatic calibration error of about 20 mm for the robotic arm. It can be seen that researching and developing intelligent and automated railway characterization defect detection equipment systems and methods is currently an important task.

At the same time, with the development of intelligent transportation technology, it is particularly important to ensure the service life and quality of high-speed train tracks. This requires the real-time monitoring and detection of the service life and quality of tracks, whether under conventional conditions or extremely harsh conditions. This will inevitably bring great difficulties to track inspection, increase the cost and difficulty of intelligent rail inspection, and further highlight the importance of intelligent, digital, and comprehensive rail quality inspection and monitoring equipment and methods.

## 3. Non-Destructive Testing Methods for High-Speed Train Rails

### 3.1. Ultrasonic Waveguide Testing Methods

The detection of surface defects on railway tracks based on ultrasonic waveguide technology is currently widely used in the field of high-speed train rail surface defect detection. With the development of intelligent transportation technology, the requirements for the service life and quality of railway tracks are becoming increasingly high. At the same time, various types of surface quality defects, such as dents, cracks, surface burns, weld defects, joint defects, etc., have also occurred during the service of railway tracks. The most important components in the ultrasonic waveguide detection process are the linear ultrasonic head and the angular ultrasonic head [28]. At the same time, different angles are arranged according to the layout of the railway track, with the aim of covering relevant areas in different regions of the railway track. In Germany, two linear ultrasonic heads emitting longitudinal waves are used to simultaneously detect contact surfaces running parallel to the railway tracks, and a matching reflector is used to detect the height of the railway tracks. One of the two ultrasonic heads adopts a dual element setup, installed on the rail head and reflector near the contact surface of the rail, while the other ultrasonic head is installed on the rail web and rail foot. Normally, the installation covers an angle range from +35 ° to +70 ° and from −35 ° to −70 °, as shown below in Figure 6.

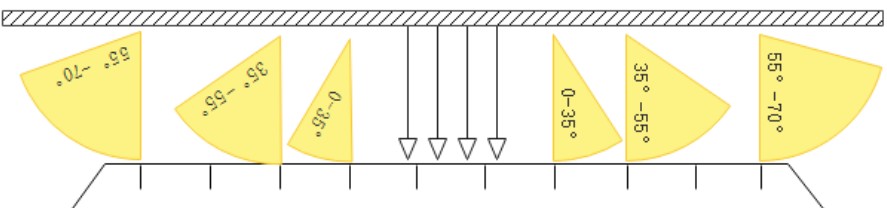

**Figure 6.** Schematic diagram of ultrasonic head installation.

By using ultrasonic sensors to collect the characteristics of rail defects, the quality and accuracy of the surface defect feature map of the rail are relatively high, and the acquisition

efficiency is high. However, in order to obtain complete and high-quality rail defect feature images [29], the ultrasonic head still needs to be installed at different angles. Shown in Figure 7 is a schematic diagram of ultrasonic wave acquisition at different angles.

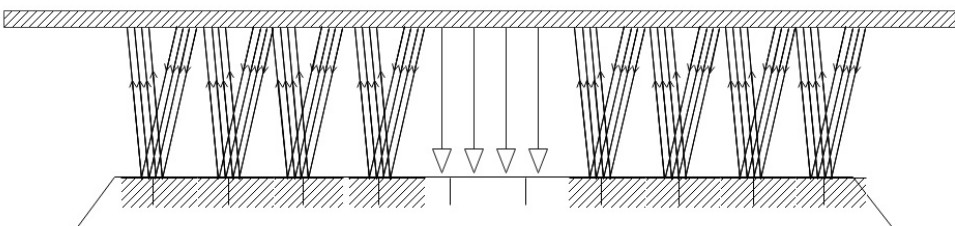

**Figure 7.** Ultrasonic wave acquisition schematic diagram at different angles of ultrasonic head.

Electromagnetic ultrasonic testing technology excites and receives ultrasonic waves through the principle of electromagnetic coupling. A coil with alternating current is placed near the surface of the rail to be tested. Under the influence of the alternating current, eddy currents are generated inside the rail. The eddy currents inside the rail are affected by the Lorentz force under the action of the alternating magnetic field, generating ultrasonic waves inside the rail. Compared with traditional ultrasonic testing techniques, electromagnetic ultrasound has the characteristics of non-contact implementation, no need for coupling agents, and fast detection speed [30,31]. Currently, Rose J.L. et al. in the United States have used electromagnetic ultrasound testing technology to achieve damage detection in steel rails [32]. Xu Ji et al. [33] in China studied a digital signal processing method suitable for the electromagnetic ultrasonic testing of rail treads. This method can improve the signal-to-noise ratio of electromagnetic ultrasonic testing signals and achieve the localization and imaging of rail tread damage. Compared with the disadvantage of traditional ultrasonic detection technology being susceptible to external interference, laser ultrasonic detection technology can adapt to complex and harsh detection environments with high detection accuracy [34]. Cavuto [35] et al. developed a non-contact air coupled laser ultrasonic detection system, demonstrating the applicability of laser ultrasonic programs in improving the ultrasonic detection performance of train axles, and it has been applied to the inspection of high-speed train axles by Italian railway companies. Sun Jihua et al. [36] applied a laser electromagnetic ultrasound technology to defect detection in the rail waist area which can detect through-hole defects of 6 mm~8 mm in the rail waist area. The study by scientist Rayleigh [37] on the propagation of elastic waves on solid surfaces is considered the beginning of guided wave research, and these elastic waves are named Rayleigh waves. Horace Lamb [38] studied elastic waves inside a flat plate based on Rayleigh's research, derived the wave equation of elastic waves in a free state plate structure, obtained Lamb waves by solving the wave equation, and expressed the dispersion characteristics of the elastic waves using this wave equation, making a historic contribution to the development of guided waves. In 1959, Gazis [39,40] studied the propagation of guided waves on the surface of tubular structures. By using Helmholtz decomposition to obtain arbitrary values of the physical parameters involved, characteristic equations applicable to hollow cylindrical structures were obtained, and complete analytical expressions for axisymmetric and non-axisymmetric guided waves were derived.

In the research and design of ultrasonic guided wave excitation and reception devices, the commonly used excitation and reception guided wave methods are divided into three types: piezoelectric, electromagnetic, and laser pulse. Loveday P.W. et al. [41] studied a guided wave excitation sensor that can be used for rail detection and verified the feasibility of using piezoelectric transducers in rail detection through simulation analysis and experi-

mental verification. Rose J.L. et al. [32] used non-contact air coupling and electromagnetic ultrasonic transducers (EMATs) to excite and receive ultrasonic guided waves at the rail head and rail waist and effectively detect the structural state of the rail. Rizzo P. et al. [42] designed a transducer applied to the rail head and collected and analyzed the sensing signals received by the rail head. Lu [43] and Loveday [41] designed and optimized, respectively, electromagnetic ultrasonic guided wave transducers for damage detection in steel rail bottoms.

The phase velocity of a guided wave is the propagation velocity of a point with a fixed phase in the direction of its propagation, while the group velocity of a guided wave is the propagation velocity of an envelope composed of waves with similar frequencies. The group velocity represents the propagation velocity of the wave group. The general definitions of group velocity and phase velocity are

$$
\begin{aligned}
C_p &= \frac{\omega}{k} \\
C_g &= \frac{dw}{dk}
\end{aligned}
\tag{1}
$$

Among them, $\omega$ is the angular frequency, $k = \frac{2\pi}{\lambda}$ is the wavenumber, and $\lambda$ is the wavelength. Further deformation results in the relationship between group velocity and phase velocity, expressed as

$$
C_g = \frac{C_p^2}{C_p - \omega \frac{dC_p}{d\omega}}
\tag{2}
$$

In the process of rail damage detection, the commonly used method for solving the dispersion curve is the potential function method, which is implemented with the particle motion equation derived by the Lame–Navier equation [42,44]:

$$
\mu \cdot u_{i,jj} + (\lambda + \mu) \cdot u_{j,ji} + \rho \cdot f_i = \rho \cdot \ddot{u}_i (i, j = 1, 2, 3 \ldots)
\tag{3}
$$

Among them, $\lambda$ and $\mu$ are Lame Changshu parameters, $\rho$ is the density, $u$ is a unique particle, and $f_i$ is the force acting on the particle.

Although ultrasound can achieve the identification and localization of rail damage, there are still certain limitations in the refinement of damage types and quantitative analysis. The detection speed reaches 15 km per hour, and the detection rate can reach 91%. There is still significant research to be conducted in the field of the intelligent identification and detection of rail damage.

### 3.2. Terahertz Detection Methods

Terahertz (THz) radiation typically refers to electromagnetic waves with frequencies between 0.1 THz and 10 THz (wavelengths between 30 μm and 3 mm), which fall between the microwave and infrared wavelengths and belong to the far-infrared band. For the online real-time detection of surface defects on high-speed train tracks, terahertz time-domain spectroscopy systems can achieve the detection of quality defects on the track surface. They mainly include a terahertz time-domain spectroscopy system based on the femtosecond laser excitation of photoconductive antennas. The terahertz optical path system is composed of a fiber-coupled terahertz transmitter, a fiber-coupled terahertz receiver, a 100 V high-voltage generator, a fiber-coupled delay line, a high-speed data acquisition board, a portable computer, an external lens mounting bracket and a TPX lens, control software, and an acrylic protective plate. The overall optoelectronic scheme of the whole fiber terahertz time-domain spectroscopy system is shown below in Figure 8.

When terahertz waves are incident at a certain angle and come into contact with the surface of the steel rail through the sample, they are completely reflected, resulting in the total reflection of terahertz waves by the steel rail. The thickness detection of THz TDS using the reflective THz method involves the incident terahertz wave E (w) on a rail

damage feature with a thickness of *d*. Eper is emitted from the upper surface of the rail damage feature, with an arrival time denoted by T. E is emitted from below the sample, with an arrival time denoted by T. The thickness of the sample can be expressed as [45,46]

$$d = \frac{c\Delta T}{2\sqrt{n^2 - \sin\theta}} = \frac{c(T_2 - T_1)}{2\sqrt{n^2 - \sin\theta}} \tag{4}$$

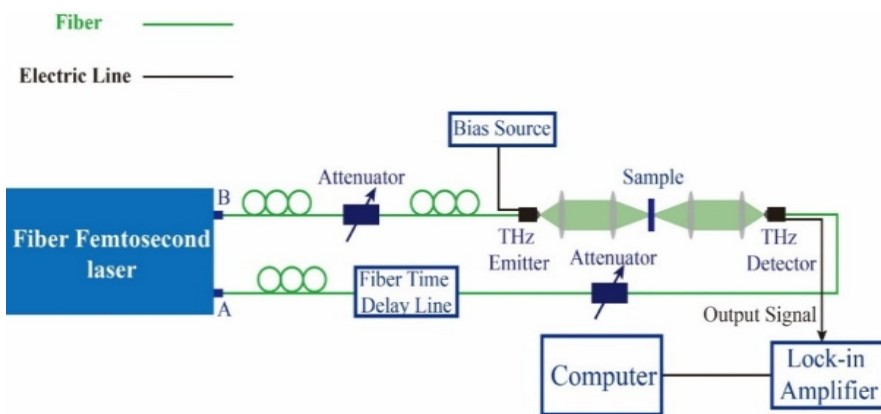

**Figure 8.** Overall structure of terahertz time-domain spectroscopy system.

Among them, *c* is the speed of light waves, and *n* is the refractive index of the material.

When terahertz waves are incident at a certain angle, the THz waves reflected by the sample cannot be completely received by the terahertz receiver. To avoid this situation and ensure that terahertz waves are vertically incident on the sample, the formula can be simplified as

$$d = \frac{c\Delta T}{2n} = \frac{c(T_2 - T_1)}{2n}$$

For the internal defects of laminated steel rails, non-destructive testing is carried out using a terahertz time-domain system. The only improvements made are in signal and image processing, and the results obtained are not yet perfect. Therefore, it is necessary to start with the hardware of the terahertz system to break the limitations of the terahertz system itself and improve the quality of signals and images. At present, terahertz systems are bulky and only remain in laboratories, making it difficult to perform on-site detection. Under ideal conditions (such as a clean rail surface and a defect size $\geq 0.1$ mm), the detection rate of surface open cracks using terahertz time-domain spectroscopy (THz TDS) or terahertz imaging technology can reach over 90%.

Therefore, it is necessary to further optimize and improve various devices and processes to truly achieve the miniaturization of terahertz systems and meet the requirements of on-site detection. With the development of deep learning, algorithms for detecting and classifying defects also need to be developed accordingly; otherwise, sometimes, defects may be missed or incorrectly detected due to improper algorithms.

### 3.3. New Eddy Current Testing Methods

At present, many foreign scholars use eddy current pulse thermal imaging technology to detect metal cracks, and their research results also prove the feasibility of eddy current pulse thermal imaging non-destructive testing technology for metal crack detection [47,48]. Robin Clark summarized the defect detection techniques currently used for rolling contact fatigue cracks in railway rails and also stated that pulse eddy current thermal imaging non-destructive testing technology will be widely applied to the detection of surface and subsurface defects in rails [6]. Wilson et al. used pulsed eddy currents to detect surface and subsurface defects in steel rails and pointed out that processing time-domain and frequency-

domain signals can improve the efficiency of defect detection in the detection system [49]. Beata conducted a study on the thermal response of surface cracks in magnetic materials and found that crack depth has an impact on the thermal response. Simulation analysis was conducted, and a formula for calculating crack depth was ultimately derived [50]. Thomas et al. found that eddy current pulse detection technology can identify two different types of defects at the same location and verified it using simulation analysis [51]. Wilson et al. proposed that boundary effects affect the detection efficiency of eddy current pulse detection and proposed using normalized input signals to improve detection efficiency. The results showed that this method can improve the detection efficiency of defect areas in specimens [49]. Liu Zewei et al. used an eddy current pulse thermal imaging detection method to detect small fatigue cracks on metal surfaces and designed a non-destructive testing system [52].

With the development of technology, higher requirements have been put forward for the capacity of modern transportation, and at the same time, the load-bearing capacity of railway tracks has also increased. Under high-speed and high-load conditions, the stress on the railway track is also greater, and so is the damage to the railway track contact surface caused by the contact between the wheel and the rail. The main type of defect is the contact fatigue crack on the outer side of the railway track surface, also known as the rail head crack. In response to such defects in railway tracks, BAM [29] has collaborated with German [53] railways and other partners to develop a new eddy current measurement system that can perform real-time detection in the rail surface to determine the degree of rail damage and perform effective maintenance (Figure 9).

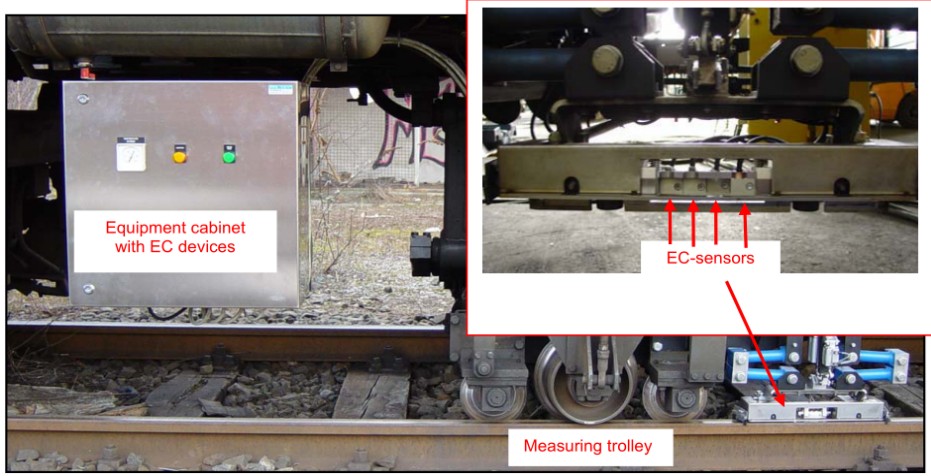

**Figure 9.** New eddy current detection system.

The new eddy current testing car can achieve real-time recording, real-time data processing, and the analysis of surface defects on rails. It can simultaneously collect and analyze eddy current signals from eight probes in real time. Based on the real-time-collected rail surface defect data, it can select, convert, and provide real-time feedback on the strongest eddy current signal [54], including the depth, location, and remaining number of eddy current head inspections of rail surface defects, as well as the continuous availability of its data (Figure 10).

Changhang et al. applied eddy current pulse thermal imaging non-destructive testing technology to defect detection in composite materials, and the experimental results proved that this testing technology provides a new idea for defect detection in non-conductive materials [55]. Eddy current pulse thermal imaging non-destructive testing technology is a relatively efficient and fast active non-destructive testing technology. Eddy current pulse thermal imaging non-destructive testing technology uses the principle of electromagnetic

induction to induce eddy currents on the surface of the test piece and then uses the principle of induction heating to achieve temperature differences between the defective and non-defective areas on the surface or subsurface of the test piece. Finally, infrared images are obtained through an infrared thermal imaging instrument to achieve defect detection. The process of the induction heating of the test piece can be roughly divided into three parts: when a high-frequency excitation current is applied to the induction coil, induced eddy currents will be generated on the surface of the test piece; after inducing eddy currents on the surface of the specimen, a large amount of Joule heat will be generated; Joule heating in defective areas can be achieved through thermal conduction and other methods to heat non-defective areas.

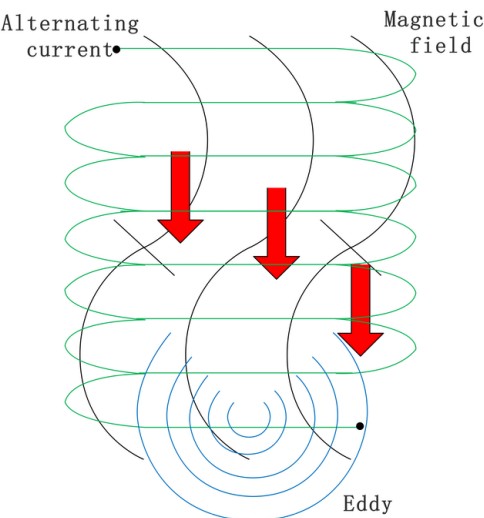

**Figure 10.** Principle diagram of new eddy current testing.

For surface open cracks with a length ≥ 0.5 mm and a depth ≥ 0.1 mm, the detection rate can reach 90~95%. When a high-frequency alternating current with an excitation frequency of f is applied to the excitation coil, an induced current will be generated on the surface of the test piece according to the principle of electromagnetic induction. At this point, the physical relationship between magnetic and electric fields can be clearly explained through Maxwell's equations, so it is necessary to focus on introducing Maxwell's equations here. The differential form of Maxwell's equations is as follows:

$$\nabla{\cdot}\overline{D} = \rho$$
$$\nabla{\cdot}\overline{B} = 0$$
$$\nabla{\cdot}\overline{E} = -\frac{\vartheta\overline{B}}{\vartheta T} \tag{5}$$
$$\nabla{\cdot}\overline{H} = \overline{J} + \frac{\vartheta\overline{D}}{\vartheta T}$$

Among them, $B$ is the magnetic flux, $H$ is the magnetic field strength, $E$ is the electric field strength, $J$ is the current density, and $\rho$ is the charge density.

### 3.4. Laser Ranging Detection Methods

Laser rangefinders typically operate based on the principle of laser propagation time. The laser emitter emits a laser beam towards the surface of the rail and measures the time required for the pulse to reflect off the rail surface and return to the sensor. The measurement principle is shown in Figure 11. At the same time, corresponding laser heads and laser emitters can be arranged according to the arbitrary shape of the rail, such as longitudinal horizontal, alignment, transverse horizontal, and torsion of the rail. Multiple laser rangefinders can be used for measurement, which is called the chord

method. Figure 11 shows the principle of the chord method for measuring the longitudinal horizontal length of the rail. This method usually uses the distance between the front bogie and the rare bogie as the reference chord and measures the main vector at 10 midpoints [56]. Extracting specific objects plays an important role in laser ranging data processing.

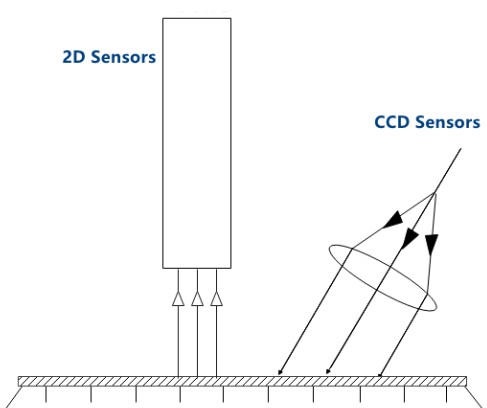

**Figure 11.** Laser ranging detection system.

Wuhan University has developed a system for measuring the external dimensions of steel rails using laser ranging sensors [57], which integrates sensor technology, laser measurement technology, multi-sensor integration and control technology, computer software technology, and network communication technology. Laser ranging sensors belong to a type of point structured light, which utilizes advanced laser triangulation technology to achieve fast measurement response speed and high measurement accuracy.

In terms of the detection principle, the line structured light measurement method uses optical technology for 3D measurement, which has the advantages of non-contact deployment, fast speed, simple implementation, high accuracy, and good anti-interference ability and currently makes it one of the most widely used methods in 3D measurement. The point structured light rail shape measurement method developed by Wuhan University has been applied to Wuhan Welding Rail Factory and has achieved certain results, confirming the superiority of optical technology in rail shape detection. Line structured light is superior to point-by-point scanning and slow-point structured light while avoiding the complex algorithm of encoding and decoding required for multiline structured light, making it more suitable for the development of this system. Using a pulse laser rangefinder, the ranging accuracy of the rail surface can reach $\pm 0.05 \sim 0.1$ mm. A 1550 nm fiber laser is used in conjunction with a high-precision time measurement chip.

Laser sensors are currently highly mature products in the industrial sector, widely used in machine tools, production lines, the rubber industry, and the steel industry. Their measurement accuracy can reach 0.019 mm. At the same time, the displacement sensor accuracy is 0.015 mm, and the guide rail accuracy is 0.012 mm, ensuring the accuracy of the system in all aspects.

*3.5. LiDAR Detection Methods*

In the research and application of non-destructive testing for surface defects on railway tracks, in addition to the laser ranging detection method, another method is to use the laser ground-penetrating radar (GPR) detection method, which mainly uses a radio wave source to generate electromagnetic energy pulse waves on the detected railway track surface (Figure 12). The electromagnetic energy pulse waves are transmitted in different material media, accompanied by reflection between contact surfaces, and finally, the reflected energy is received and analyzed [58]. Therefore, laser ground-penetrating radar can map the surface structure of railway tracks and the underground conditions of railway track structures.

The collected rail surface data information is processed through relevant algorithms and serialized to achieve the non-destructive testing of rail surface quality. The commonly used processing algorithms include machine learning algorithms, such as artificial neural network (ANN) algorithms and CNNs [59].

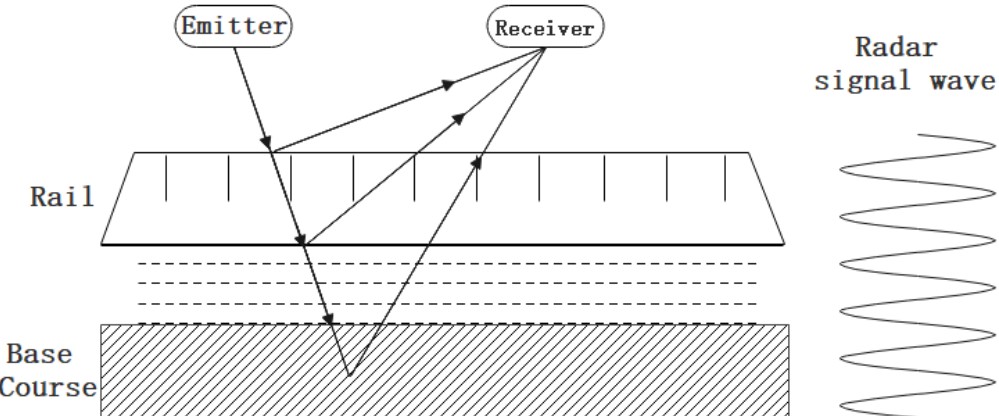

**Figure 12.** Laser radar detection method.

The clustering algorithm is commonly used in the field of LiDAR obstacle detection. Its principle is to divide the sample into multiple categories, with the similarity between points within the same category as high as possible and the similarity between different categories as low as possible, in order to reveal the distribution of data. The commonly used clustering algorithms include Density-Based Spatial Clustering of Applications with Noise (DBSCAN), the K-means clustering algorithm, and the Euclidean clustering algorithm. The DBSCAN algorithm is a clustering algorithm based on data density features which can cluster point cloud data of any shape. Cai Huaiyu et al. improved the traditional DBSCAN algorithm by setting an adaptive clustering neighborhood radius that varies according to the target distance, which improved the accuracy of obstacle detection [60]. Zhang Changyong et al. proposed an adaptive DBSCAN algorithm, which selects representative core points and adaptive clustering radii to achieve the fast and accurate clustering of obstacles with different densities [61]. Chen et al. designed the coplanar threshold as an additional clustering condition, where the clustering threshold is automatically adjusted to adapt to the local distribution of samples in the input dataset without the need to adjust the parameters [62].

The method based on a ground grid usually involves projecting a 3D point cloud onto a 2.5D grid map. A 2.5D grid map, also known as an elevation map, stores the height information of point clouds. Douillard et al. extended the elevation map by combining the average height within the grid and the voxel-based object model of the point cloud, and the resulting model is also applicable to path planning [63]. Shao Jingtao et al. first clustered the three-dimensional point cloud into grids based on gradients and finally used cubic B-spline curves for smooth fitting and the segmentation of the ground. Himmelsbach et al. used local geometric features of three-dimensional data to segment the ground and objects, achieving good segmentation results in urban environments including smooth and curved road surfaces [64]. The method based on plane fitting usually utilizes the three-dimensional information of the original point cloud data to construct a ground model and distinguishes ground points based on the degree of matching between each point and the model. Himmelsbach et al. proposed an extended Gaussian regression method to fit the ground, utilizing non-stationary covariance functions to locally adapt to terrain data structures [65]. Plagemann et al. proposed a probabilistic technique based on Gaussian processes which more reliably predicts the elevation of unseen locations than

other methods while estimating the uncertainty of predictions [66]. Chen et al. used a one-dimensional Gaussian process (GP) with non-stationary covariance functions to distinguish ground points or obstacles in each segment, decomposing large-scale ground segmentation problems into many simple GP regression problems [67]. Zermas, Guan Junzhi et al. used the Random Sample Consensus (RANSAC) algorithm to fit a plane model to segment the ground and fitted the ground plane into segments. For each segment, RANSAC was used to extract the plane, effectively overcoming the problem of poor ground segmentation [68,69]. In the classification and recognition of point cloud objects, it is necessary to provide a dataset with good performance in classification. However, the dataset only contains three types of objects, cardboard boxes, pedestrians, and trains, and the number of object types is relatively small. In future research, the dataset can be expanded to further validate and improve the research method.

### 3.6. Laser Vision Inspection Methods

The railway track structure is not a large-area planar structure, so laser vision technology is often used to detect surface defects on the railway track based on visual technology. A laser vision system combines structural light sources with image acquisition devices to capture 3D images of railway track surface features. In order to efficiently and effectively capture railway track surface feature images, multiple angle acquisition devices are used to coordinate with each other (Figure 13). For example, one acquisition device is responsible for the inner feature image of the railway track, and another acquisition device is responsible for the outer feature image of the railway track.

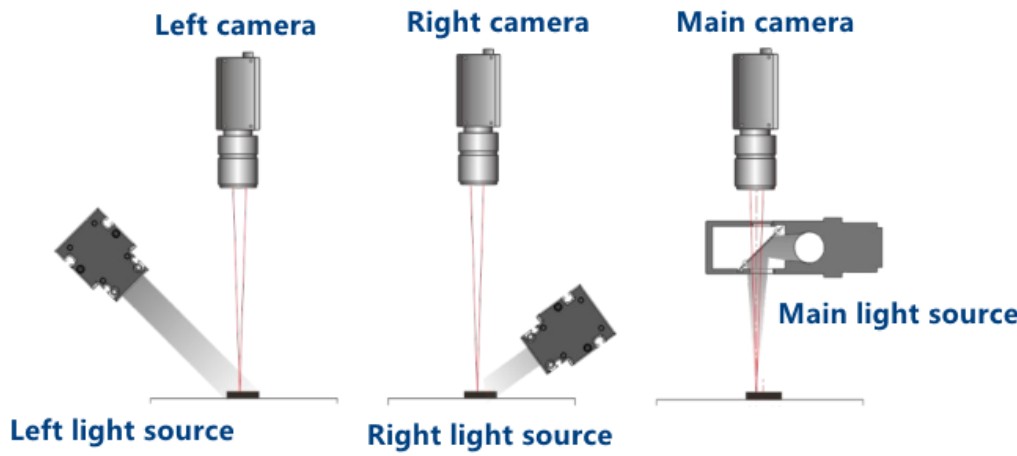

**Figure 13.** Schematic diagram of laser vision inspection.

Through this method, real-time, efficient, and high-quality image data of rail surface features can be obtained, providing data for the detection of rail surface wear and rolling contact fatigue [70]. The laser vision system can also be combined with an inertial motion unit (IMU) to collect and measure a series of geometric parameters of the railway tracks in [71]. For example, the detection process of rail corrugation includes extracting effective laser stripe areas, suppressing stripe highlight textures, extracting stripe centerlines, defining rail boundaries, and measuring the position of and deviation in laser measurement points, as shown in Figure 14 [72]. Of course, gratings of different wavelengths can also be used to capture surface feature image data of railway tracks through illumination.

In 2002, You Yang, Liu Ming, Cui Chunyan, and others from Hebei University applied the light cutting method combined with visual inspection technology in the detection of rail wear, greatly reducing the time required for rail image processing [73]. In 2014, Zhan Dong, Yu Long, and others from Southwest Jiaotong University proposed a detection method for

rail wear cross-sectional area based on existing rail inspection vehicles. They discretized the analytical formula for the standard rail profile and calculated the step integration based on profile registration to obtain the calculation formula for rail wear cross-sectional area [74]. Xiao Longfei, Li Jinlong, Gao Xiaorong, and others from Southwest Jiaotong University, in 2015, proposed to reconstruct and restore the three-dimensional spatial contour of steel rails based on phase measurement contour technology. The research results showed that the online measurement of the three-dimensional contour of steel rails can be achieved. This method has good application prospects in measuring rail wear [75].

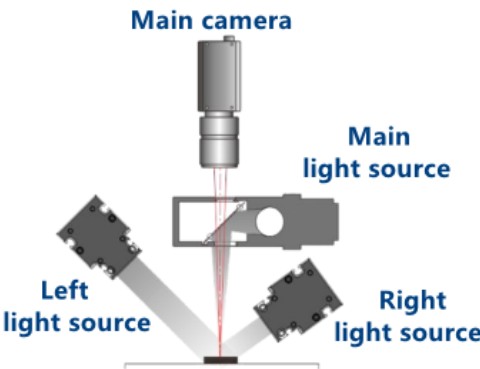

**Figure 14.** Multiple-light-source layout for laser visual detection.

Wang Jiwu, Zhang Xianwen, Gao Weijie, and others from Beijing Jiaotong University conducted in-depth research and analysis on the calibration method and accuracy impact of cameras in measurement systems. They proposed a spatial coordinate conversion algorithm for laser contour points and applied it to develop a non-contact dynamic measurement device for rail wear, with an accuracy of 0.012 mm and a relative error of 0.28% [76]. In 2015, Xu et al. combined laser vision measurement technology to reconstruct the full contour of the rail using the condition that the left and right rail head treads have the same characteristics, combined with the iterative nearest-point algorithm for accurate matching. They also achieved the dynamic measurement of rail wear within a certain detection distance. The performed tests on a manually operated track inspection vehicle at a track site. The results show that the calibration accuracy of the measurement system is 0.0049 mm, and the wear measurement accuracy can reach 0.1 mm [77].

### 3.7. Infrared Thermal Sensing Detection Methods

Infrared thermal sensor technology is widely used not only in the military field but also in the field of industrial product quality inspection. Infrared thermal sensor technology mainly relies on thermal sensor imaging to obtain thermal images of the detection object; then, based on a series of processing and analysis algorithms developed, it is processed and analyzed to obtain results. Due to the contact operation between the wheel and the rail in high-load and high-speed scenarios, a significant amount of heat is generated on the surface of the rail. The use of infrared thermal bed technology can effectively capture thermal images of the rail surface features, thereby enabling the real-time detection of rail surface defects. At the same time, in the processing of infrared thermal images of railway tracks, there are a series of processing algorithms that can accurately identify and detect surface defects, including damage and defects inside the railway tracks [78]. Infrared thermal sensing detection technology is based on infrared radiation for detection, which is a type of electromagnetic radiation. Infrared radiation is divided into four wavelength bands, as shown in Figure 15: near-infrared (NIR), 0.78–3 um; mid-infrared (MIB), 3–6 um; far infrared (FIR), 6–15 um; and extremely far infrared (XIR), 15–1000 um.

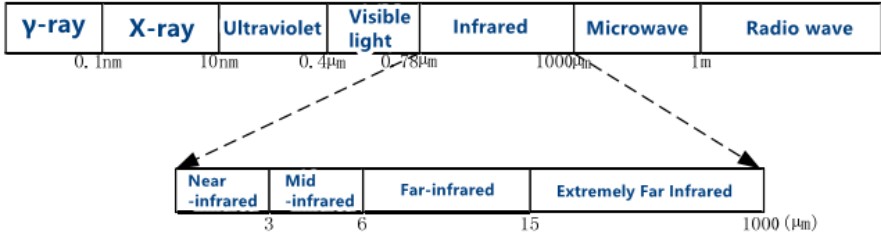

**Figure 15.** Infrared heat sensing wave map.

With the continuous development of infrared technology, people have increasingly high requirements for the clarity of infrared images, and many new algorithms have been proposed abroad for infrared image processing. In 2011, scholar H. Gökhan İlk proposed an adaptive Laplacian filtering algorithm for infrared image enhancement. This algorithm achieves image edge sharpening by minimizing the objective function, improving the clarity and contrast of infrared images [79]. In 2013, American scholar Kelly designed an infrared image detection system based on pixel focal plane array which improved the imaging capability of infrared images [80]. In 2015, Malaysian scholar Yuan T. proposed an adaptive contrast enhancement algorithm for infrared image enhancement which improves the visual quality of infrared images by increasing the difference in contrast between the target and the background [81]. In 2016, Indian scholar Lincelles et al. proposed an infrared image enhancement algorithm based on multimodal image fusion which combines multi-scale decomposition and principal component analysis to improve contrast in infrared images [82]. In 2010, Finnish scholar Bochko proposed a manual correction infrared image segmentation algorithm which can analyze and measure the surface area of lesions in human infrared images after segmentation processing and observe their changes over time, which is helpful for clinical diagnosis and treatment [83]. In 2012, American scholar Jadin applied infrared image segmentation technology to power equipment detection, extracting the thermal distribution of electrical equipment from segmented images [84]. In 2015, Indian scholar Mangai proposed an infrared image segmentation algorithm based on K-means clustering which uses the concepts of fitness and belonging to make the results of image segmentation more applicable [85].

Infrared thermal wave non-destructive testing technology mainly includes three aspects: heating technology, thermal imaging technology, and thermal image processing technology. It can be combined with other detection methods to improve the accuracy of object detection. Compared with traditional non-destructive testing methods such as electromagnetic and eddy current, infrared thermal wave non-destructive testing has the advantages of wide applicability, fast detection speed, large detection area, one-way non-contact detection, and quantitative detection. This article mainly introduces the principle and detection methods of infrared thermal wave non-destructive testing, as well as the application of infrared thermal wave non-destructive testing technology. Thermal waves are temperature fields that vary over time. When any object in nature is disturbed by external factors, its temperature field changes. Due to the unevenness of the material and internal structure of the object, the internal heat conduction process of the object is different, and different heat conduction velocities result in different temperature distributions on the surface of the object. The transmission process of thermal waves, like that of any wave, has its own specific transmission rules. During the transmission process, thermal waves interact with the material and internal structure of the object. The theory of thermal waves aims to study this interaction and analyze the transmission process of thermal waves inside objects. The Fourier law proposed by French physicist Fourier in 1822 is a fundamental law in the field of heat transfer, revealing the relationship between heat flow rate and temperature gradient. Its expression for the differential equation of thermal conductivity is

$$q(x,t) = -\lambda \nabla T(x,t) \tag{6}$$

Among them, $q(x,t)$ is the heat flux vector, $\nabla T(x,t)$ is the temperature gradient, $T(x,t)$ is the instantaneous temperature, and $\lambda$ is the thermal conductivity.

### 3.8. Artificial Intelligence Detection Algorithms

At present, in practical applications, including digital twin technology [86], traditional detection methods detect the surface defects of steel rails through manual visual inspection, rail inspection vehicles, physical methods, etc. The main problem is that during the process of manually inspecting or scanning the surface images of steel rails by rail inspection vehicles, complex and changing natural environmental conditions such as background conditions, light, and weather can interfere with the detection process to a certain extent, resulting in low image quality and subsequently affecting the accuracy of detection [87,88].

When traditional methods cannot meet the requirements of automation and intelligence, convolutional neural networks are applied to track surface damage detection, opening up a new development path for detection work. To effectively compensate for the shortcomings of traditional image processing methods, many researchers suggest using deep learning methods to design fast and accurate track surface damage detection algorithms. Jin Xiating et al. [89] improved the convolutional neural network by combining attention mechanisms, focusing the model on the defect areas on the surface of the rail. Using semantic segmentation methods, they achieved high segmentation accuracy for the contour segmentation of rail surface defects. Joquab et al. [90] used classical networks as the backbone extraction network and employed transfer learning methods to extract the surface defect features of steel rails. They designed adaptive edge and spatial feature extraction structures in the network to enhance the model's expressive power. Su et al. [91] used the Faster R-CNN object detection method to perform damage detection on B-type images of internal track damage. Although this method can effectively identify surface damage, its detection ability is single-focused on track damage due to the difficulty in obtaining track defect datasets. Zhao et al. [92] conducted defect detection on key components of railway tracks, which was mainly divided into two parts. Firstly, the position of the key components was recognized, and based on the recognized images, defect classification detection was carried out. The detection task was completed by segmenting the surface defect images of the steel rails using super-resolution algorithms.

Zeiler et al. [93] used deconvolution to reconstruct images and improve the classification performance of the model. The results showed that the surface reconstruction method was very effective in defect detection. Wang et al. [94] used convolutional neural networks based on array methods to achieve the damage detection of rail scars and cracks. Liu et al. [95] proposed the Wav2Vec neural network for piezoelectric array ultrasonic guided wave localization based on ultrasonic guided wave SHM technology, achieving the accurate localization of rail damage and building an ultrasonic guided wave detection system. Zeng et al. [96] used fiber Bragg grating sensing technology to monitor rail damage and obtain data, addressing the problem that traditional high-speed railway sensors cannot cope with electromagnetic interference. They then used convolutional neural networks to detect the damage [97]. Considering the large number of rail point cloud data obtained through on-site scanning and the presence of a large number of noise points and outliers, a point cloud simplification algorithm was proposed to detect rails through direct filtering, maximizing the acquisition of detailed information about the rails. Zhang Xiaoyu et al. [98] proposed a rail top defect detection method based on cascaded networks, targeting the characteristics of rail top defects. On this basis, deep learning algorithms were added to detect defects on the rail top surface. To address the drawbacks of traditional image processing methods such as long processing time and low defect recognition accuracy,

a new convolution kernel combined with a cascade-based feature fusion strategy was chosen to design an improved Faster R-CNN rail defect detection method for implementing detection tasks [99].

*3.9. Multi-Sensor Fusion Methods*

Multi-sensor data fusion and decision fusion of multiple classifiers both belong to the category of information fusion, which originated from the sonar signal processing system funded by the US Department of Defense in 1973. At present, the level of information fusion can be divided into low level (data level or pixel level), middle level (feature level), and high level (decision level) based on the different levels of fusion objects, which facilitates the description of fusion behavior at all levels according to the fusion task.

Research on data fusion technology in China started relatively late, and it was not until the late 1980s that reports on multi-sensor data fusion technology began to emerge. Among them, An Xueli et al. studied multi-source features based on vibration signals in the time domain, frequency domain, and envelope spectrum and used decision fusion methods to construct a wind turbine bearing fault diagnosis model [100]. Gao Jingwei et al. studied a fault diagnosis method for self-propelled artillery gearboxes based on information fusion [101]. Jiang Wanlu et al. studied the information fusion fault diagnosis method of Bayesian networks [102]. Zhai Xusheng et al. studied a multi-sensor information fusion method for diagnosing vibration faults in aircraft engines [103].

The theory of data fusion has been widely applied in numerous military and civilian fields, such as industrial control, robotics and intelligent instruments, air traffic control, ocean surveillance and management, and more. In these fields, the fundamental theoretical research of data fusion, the construction of a data fusion theoretical system, and the study and establishment of a basic theoretical framework for multi-sensor integration and information fusion are some of the research directions. In addition, for database management, introducing various artificial intelligence technologies into the field of data fusion, reducing computational complexity in the context of heterogeneous sensors on multiple platforms, and solving the expression and inference operations of uncertain factors are also key areas of future research. Relatively speaking, there are few examples of applying multi-sensor data fusion technology to rail track detection. Some scholars have proposed using machine learning methods to detect wheel crack states using multiple sensors [104], while others have proposed the idea of using multiple sensors to detect obstacles in railways [105]. Finding a detection method for rail cracks is gradually becoming an increasingly urgent need, and this project is based on the multi-sensor measurement of acoustic emission signals to detect rail cracks, which currently has great development prospects.

## 4. Railway Non-Destructive Testing Platforms and Equipment

In the field of transportation, the safety and reliability of railway tracks play a crucial role in the entire transportation system. And some instruments and equipment used for reliability fault prediction and rail defect detection in the early stage have also become hot research topics in recent years. Based on the above summarized railway track detection technology and methods, related railway track detection platforms, railway track detection robots, and railway track detection equipment have been developed and have a certain range of applications.

*4.1. Rail Inspection Robot System*

Faced with the characteristics of multiple categories, types, and features of surface defects on railway tracks, the railway track detection robot system based on 2D vision technology [106] has become a research hotspot. The main method is to collect surface

feature images of railway tracks through image acquisition sensors, perform relevant algorithm feature analysis and processing on the railway track feature images, and finally identify defect features and provide feedback on the results. The rail inspection robot system based on 2D vision technology mainly recognizes and detects the surface defects of railway tracks, such as cracks, pits, wear, pitting, scratches, and deformation, as shown in Figure 16.

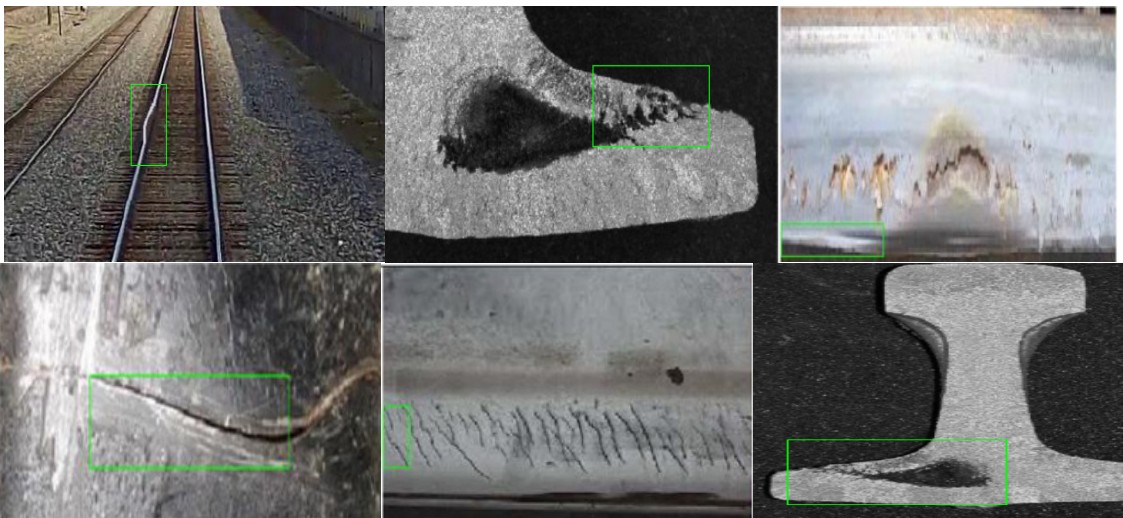

**Figure 16.** Surface defects.

The rail inspection robot collects real-time images of the rail surface through installed image acquisition sensors. The core principle is image processing. After processing the defect images of the rail surface, the processing results are fed back in real time and uploaded to the cloud in real time. The system framework diagram is shown in Figure 17.

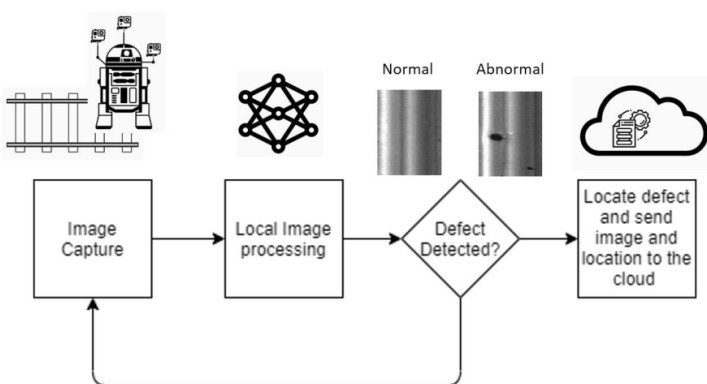

**Figure 17.** Orbit inspection robot system.

## 4.2. Hand-Pushed Rail Inspection Trolley

In order to facilitate the inspection of the surface quality of rails at different joints and of multiple-joint rails, a manually pushed rail inspection trolley is used for the inspection of rail surface quality due to its light weight, small size, and flexible operation [56], as shown in Figure 18. Due to its small size and light weight, the hand-pushed small car is very easy to install on the railway track, with low professional requirements for the operators and a low threshold for manual operation. Ultrasonic sensors, laser measurement and detection systems, and structural sensors are installed on the crossbeam of the car to detect the surface quality of the rail and transmit detection information signals. As the handcart relies on manual pushing to inspect the surface quality of the rails, the overall inspection speed

and efficiency depend entirely on the operator's travel speed and efficiency. Therefore, the inspection efficiency of this manual rail inspection trolley is not too high, and the operator needs to stop the inspection when fatigued.

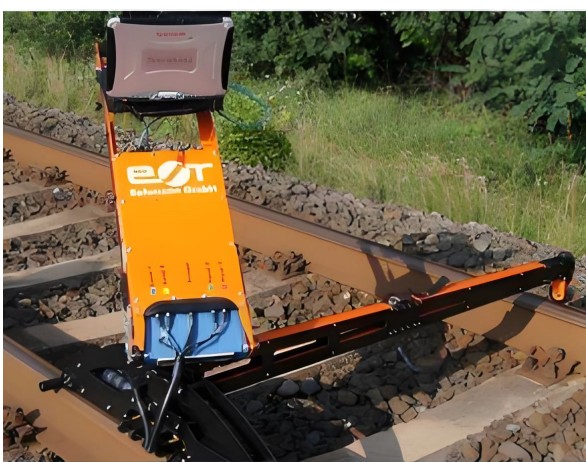

**Figure 18.** Hand-pushed rail inspection trolley.

### 4.3. Active Rail Inspection Car

The above manual rail inspection car has the advantages of light weight and small size and also has lower requirements for the operator. However, there are also some shortcomings, while the active rail inspection car [107] realizes the automation of rail inspection. The overall structure of the active rail inspection car is similar to that of the manual rail inspection car, but the most prominent feature is its ability to automatically walk and inspect the rails, as shown in Figure 19. The active rail inspection car carries instruments such as ultrasonic sensors, visual sensors, and laser sensors to actively detect the surface quality of the rail.

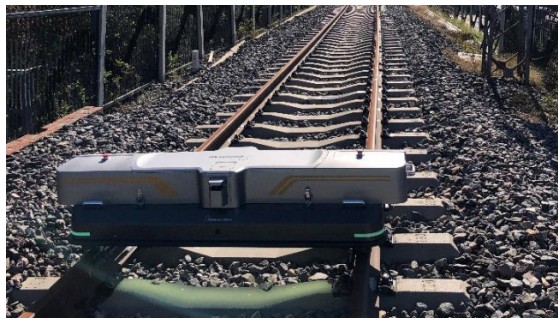

**Figure 19.** Active rail inspection car.

### 4.4. Vehicle-Mounted Rail Inspection Vehicle

The active rail inspection car overcomes the problem that the manual rail inspection car cannot actively complete the inspection. However, the active rail inspection car has limited endurance due to the lack of continuous external alignment for power supply, which greatly affects the efficiency of rail inspection. The vehicle-mounted rail inspection vehicle [108,109] effectively solves such problems. As the name suggests, the vehicle-mounted rail inspection vehicle installs relevant rail inspection instruments or sensor systems on the rail inspection car. The vehicle-mounted car can be a free-type inspection car, an improved car, or a detection train, as shown in Figure 20. A series of sensor devices can also be installed on the inspection vehicle, such as visual laser systems [110], ultrasonic sensors, eddy current sensors, EMAT [72] sensors, etc.

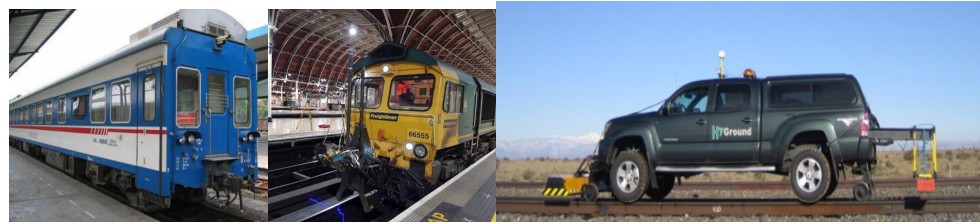

**Figure 20.** On-board rail inspection vehicle.

*4.5. Comparison and Discussion of Platform Characteristics for Four Detection Methods*

Different rail inspection methods and platform equipment can achieve the non-destructive testing of rails to a certain extent. Table 1 shows a comparative analysis of the characteristics of the summarized rail inspection methods and platform equipment.

**Table 1.** Comparison of detection characteristics.

| Serial Number | Testing Methods | Detection | Advantages | Disadvantages | Digital Twin and Life Prediction | Normal Testing Pass Rate |
|---|---|---|---|---|---|---|
| 1 | Manual detection methods | Viewing with human eyes | Direct judgment of defects | Defects are easily missed, fatigue, and low efficiency | NO | 70~85% (depending on testing experience) |
| 2 | Ultrasonic testing methods | Ultrasonic technology | Long detection distance and high accuracy | Poor noise resistance and inability to reflect defect types | NO | 85~95% (internal defects) |
| 3 | Eddy current testing methods | Eddy current effect | High detection accuracy | Low signal-to-noise ratio and low anti-interference ability | NO | 80~90% (surface/near-surface defects) |
| 4 | Laser vision inspection methods | Image recognition | Non-contact and high precision | Internal defects cannot be detected | NO | 85~95% (visible surface defects) |
| 5 | Infrared thermal sensing detection methods | Infrared thermal sensing image recognition | Non-contact and high precision | Poor noise resistance and inability to reflect defect types | NO | 70~80% (thermal anomaly defects) |
| 6 | LiDAR detection methods | Electromagnetic energy pulse detection | Non-contact and high precision | High cost and inability to reflect defect types | NO | 90~95% (geometric damage) |
| 7 | Terahertz detection methods | Terahertz time-domain spectroscopy | Non-contact and high precision | High cost and limited detection thickness | NO | 60~80% |
| 8 | Hand-pushed track inspection trolley | Manual pushing and multi-sensor fusion | Small size, light weight and easy to install | Low efficiency and slow detection speed | NO | 95~98% (comprehensive testing) |
| 9 | Active track inspection car | Automatic multi-sensor fusion | Autonomous detection and high efficiency | Low battery life | NO | 95~98% (comprehensive testing) |
| 10 | Rail inspection robot | Active multi-sensor fusion | Large detection angle and high accuracy | Unable to be controlled remotely and high cost | NO | 95~98% (comprehensive testing) |
| 11 | Vehicle-mounted track inspection vehicle | Active multi-sensor fusion | Large detection angle and high accuracy | Unable to be controlled remotely and high cost | NO | 95~98% (comprehensive testing) |

From Table 1, it can be seen that different detection methods and platform equipment have their own advantages and disadvantages. Suitable detection methods and equipment can be selected according to specific rail conditions. Electromagnetic detection methods have relatively deep development and application in theoretical research and technical application. Currently, they mainly focus on the quantitative analysis and research of rail defect detection, as well as the formulation of detection standards, including rail surface quality detection under multi-physical field coupling. However, they have certain

limitations and cannot accurately judge the rail life. Machine visual inspection methods based on image processing technology have solved the problem of low accuracy in rail surface quality inspection to a certain extent, and non-contact detection methods have also become a hot research topic. However, rail surface quality detection systems based on visual technology have poor robustness and huge computational complexity and makes it difficult to achieve high-speed rail detection, making it impossible to directly determine the rail operating status and conduct life health monitoring. Although ultrasonic testing methods can theoretically detect the surface quality of railway tracks well, there is still room for research and verification in practical applications, especially in large-scale trial and formal testing applications. Although terahertz technology has high detection accuracy in rail surface quality inspection and has certain advantages in non-contact detection and high-speed detection, there are certain limitations on the detection thickness for rail surface defects.

In recent years, a research hotspot has been rail inspection robot systems, which are undoubtedly pieces of rail inspection platform equipment with high reliability and authenticity in the field of rail inspection. They can be equipped with multiple sensors to achieve the real-time online detection of rails in all directions and angles and realize active detection with long detection distance. Rail inspection robots are currently widely used in the field of rail inspection platforms and equipment. However, there is still room for improvement in remote control, remote communication, remote interaction, and other aspects of rail inspection robot systems, as Bluetooth WIFI, local area networks, and other technologies cannot meet the requirements of remote control. At the same time, with the rapid development of modern transportation technology, digital twin technology will be a very important research method and technology in the field of rail inspection, realizing the full life-cycle detection of rail surface quality and providing accurate predictions of alignment life.

## 5. Conclusions

With the development of modern transportation technology, the quality and service life requirements for railway tracks are becoming increasingly high. Surface quality inspection and service life prediction of railway tracks have become hot and difficult research topics. Therefore, it is particularly important to adopt modern means, technologies, and methods to improve the detection of railway track surface quality. Adopting a certain technical method may allow for the effective solution of the surface quality inspection of railway tracks in certain aspects, but in railway track inspection under some characteristic working conditions, it requires that the inspection methods, techniques, and methods used be adapted to special working conditions, and some intelligent methods and technologies are preferred. Subsequent research on rail inspection will focus on the following areas.

(1) Intelligent AI track inspection robot systems. In the era of digitization, intelligence, and the Internet of Things, endowing rail inspection robots with more intelligent and digital functions is expected to be a hot and difficult research topic in the field of rail inspection robots in the future. Various sensing technologies will be more integrated, not only in the process of collecting rail characteristic data but also in the entire process of data analysis and processing, as well as in the process of interacting with humans. This fully embodies the "four layer" theory that includes the physical model layer, data layer, service layer, and application layer. The physical model layer of the rail inspection robot system and the rail inspection robot's independently constructed rail digital twin model includes geometric models, physical models, and digital models of different rails. The data layer includes physical model data, information model data, and fused data. After data preprocessing, the data fusion algorithm generates twin data. The service layer

operates in three stages, namely, the observation stage, the analysis stage, and the decision-making stage, including twin data-driven rail fault prediction and health management. The application layer includes rail health status assessment, rail fault diagnosis, fault prediction, and fault maintenance decision making. In the future, there will be more and more rail inspection robot systems with more intelligent and digital capabilities.

(2) Digital twin technology. In the modern transportation field, the quality requirements for rail tracks are becoming increasingly high. Therefore, it is crucial to be able to warn of rail faults and predict the service life of rails. Digital twin technology will be widely applied, and rail inspection technology based on digital twin technology will solve such problems. Digital twin technology can break through many physical limitations, and through simulation, prediction, monitoring, optimization, and control driven by data and models, it can achieve the continuity of rail inspection, the real-time response of detection data, and the upgrading and optimization of detection models. Based on the advantages of rail models, rail feature data, and rail data services, digital twins will undoubtedly become a hot topic in the field of rail inspection research in the future.

(3) Wireless remote interaction technology. The rail inspection vehicles, rail inspection robot systems, and vehicle-mounted rail inspection vehicles mentioned earlier can achieve wireless control and data transmission to a certain extent, but there is still room for development in long-distance wireless interaction. In the field of modern intelligent transportation, in the era of interconnected and digitized everything, rail inspection systems will be expected to have the most basic and necessary functions of wireless remote interaction. The wireless remote interaction of rail inspection systems is essentially a wireless network interconnection among people, machines, and objects. Through the comprehensive and in-depth perception of rail operation detection data, the real-time transmission and exchange of rail detection data, and the rapid analysis and modeling of data, it is the transformation and embodiment of the rail inspection system to achieve digitization, intelligence, intelligent interaction, and wireless remote intelligent interaction. In the future, rail inspection intelligent systems with wireless remote intelligent interaction will be an inevitable development trend.

(4) Real-time online status monitoring and digital and intelligent operation and maintenance of railway systems. With the deepening of research on high-speed and ultra-high-speed train technology, damage detection in track infrastructure such as roadbeds, rail fasteners, and vehicles, in addition to railway tracks, will undoubtedly be a key direction of future transportation research. How to use more digital and intelligent methods to solve real-time online rail status monitoring and ultimately achieve intelligent operation and maintenance is a key and difficult point of research. At the same time, it is necessary to consider the impact of resonance on railway tracks, vehicles, and roadbeds, as well as the analysis, processing, and feedback timeliness of real-time online status monitoring data of railway tracks, vehicles, and roadbeds. Based on this, the development of non-destructive testing methods and equipment based on intelligent algorithms for the online status monitoring of railway track damage and intelligent operation and maintenance algorithms under multi-source data signals is a key direction and a trend for the future development of high-speed and ultra-high-speed train technology.

(5) Intelligent and digital detection methods for rail damage. Domestically, unmanned aerial vehicles have been equipped with technologies such as LiDAR and AI image recognition to detect track geometry parameters and surface defects. The surface defect detection method for steel rails based on drone images has a recall rate of 93.75% and an accuracy rate of 93.6%. By combining 3DGIS and digital twin technology, air vehicle ground collaborative inspection can be achieved, significantly improving efficiency. The highest detection speed of the domestically produced GTC-80-II-J rail flaw detection car is 80 km/h, with a damage

detection rate of over 85%. Combined with 3D laser track profile measurement technology, the position of the probe wheel is dynamically adjusted, and the error is controlled at the millimeter level. The vehicle is equipped with 18 probe chips and 26 ultrasonic channels, which can detect internal cracks in the steel rails in all directions. AI and big data analysis: The 4C contact network detection system analyzes more than 10 million images through AI algorithms, and the number of defect detection operations is more than 10 times that of manual labor, with significantly better accuracy than manual labor. The intelligent detection system for railway sleepers can identify fine cracks at the 0.1 mm level and monitor production environment safety indicators in real time. The track inspection car adopts dynamic adjustment technology, with a track direction/height standard deviation of ≤0.3 mm, meeting the smoothness requirements of high-speed railways. Some railway bureaus are piloting predictive maintenance systems based on digital twins, which combine historical data to predict rail wear trends and optimize maintenance cycles.

(6) Future challenges in track damage detection. In terms of future challenges, complex environmental interference, rust, oil stains, and changes in lighting on the surface of steel rails can lead to misjudgments in image recognition. Drone detection can easily mask defect features in strong- or low-light environments. Electromagnetic interference and terrain obstruction affect sensor signal transmission, leading to a decrease in GPS positioning accuracy for mountainous railways. Real-time data collection and processing, with a single flaw detection vehicle generating several GB of data in a single inspection, requiring real-time analysis and output of results. Domestic AI chips (such as Horizon Journey 5) can achieve an inference speed of 128FPS in YOLOv5s, but there are still delays in complex models (such as 3D point cloud segmentation). The fusion of multimodal data (images, ultrasound, LiDAR, etc.) requires solving problems such as time synchronization and coordinate system calibration, which is technically difficult. Rail damage includes cracks, wear, corrosion, etc., and some defects (such as internal microcracks) are difficult to detect through a single technique. For example, ultrasound is sensitive to longitudinal cracks but has a lower detection rate for transverse cracks.

Domestic high-speed rail damage detection has initially achieved intelligence and digitization. Equipment such as flaw detection cars and drones is close to international standards in detection speed and some indicators, but there are still gaps in adaptability to complex environments, multi-sensor fusion, and AI algorithm generalization ability. In the future, it is necessary to strengthen the research and development of core technologies (such as high-precision sensors and edge computing), promote the internationalization of standards, reduce operation and maintenance costs through the "detection as a service" mode, and further improve the detection efficiency and reliability.

**Author Contributions:** Y.W.: Algorithms, calculations, writing, and experiment data processing. B.M.: Proposal of research topics and ideas, and review and editing. Y.Z.: Writing and experimental data processing. Z.H.: Data collection. S.X.: Writing and experimental data processing. All authors have read and agreed to the published version of the manuscript.

**Funding:** This research study was financially supported by National Natural Science Foundation of China [51775456], the KeyR&D projects in Sichuan Province [2023YFG0197], Self-developed Research Project of the State Key Laboratory of Traction Power [2023TPL_T08], National Key Research and Development Project (New Rail Transit System; 2024YFB4303200), and Basic Scientific Research Business Expenses of Central Universities—Special Research Project [2682022ZTPY007].

**Conflicts of Interest:** We declare that we do not have any conflicts of interest in publishing this manuscript.

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
