# Peer review of "Review on Rail Damage Detection Technologies for High-Speed Trains"

_applsci, doi:10.3390/app15147725_

Round 1

Reviewer 1 Report

Comments and Suggestions for Authors

Please find the attached document for the reviewer's comments.

Author Response

Comments 1: [In section 3 various NDT testing methods are reported but the technical descriptions are too superficial. Provide a) sufficient explanations of their working principles; b) quantitative performance benchmarks (e.g., detection depth, accuracy); c) Realworld limitations and deployment challenges.]

Response 1: [Thank you very much for the professional and valuable feedback from the reviewer. Modifications have been made as required, including an analysis of the working principles of different detection methods for rail damage, including quantitative analysis, as well as the current difficulties and future research challenges.The modified content is indicated in red font.]

Comments 2: [State-of-the-art of AI-based approaches such as deep learning or sensor fusion are mentioned briefly. Provide sufficient details to cover AI-based approaches.]

Response 2: [Thank you very much for the professional and valuable feedback from the reviewer. Modifications have been made as required, adding artificial intelligence detection methods for rail damage, including deep learning and multi-sensor fusion for rail damage detection.The modified content is indicated in red font.]

Comments 3: [The conclusion section reiterates ideas from earlier sections but lacks quantitative synthesis or critical forecasting. Statements like “more and more intelligent and digital nondestructive testing” are vague without specifying current technological gaps or research opportunities. I encourage the authors to describe in detail the gaps and challenges of current methods concerning rail damage detection.]

 Response 3: [Thank you very much for the professional and valuable feedback from the reviewer. The modifications have been made according to the requirements, and the conclusion section has been revised to supplement the specific analysis of artificial intelligence detection methods for rail damage. Currently, there are deficiencies in domestic rail damage detection methods and quantitative analysis of the gap with foreign ones.The modified content is indicated in red font.]

Comments 4: [Several sections (Sections 3.3, 3.4 and 3.5) are titled as “New eddy current testing method,” even though they describe different technologies like laser rangefinding and GPR. This appears to be a copy-paste error and suggests insufficient proofreading. I suggest the authors to carefully profread the entire article.]

 Response 4: [Thank you very much for the professional and valuable feedback from the reviewer.The modifications have been made, and the font color in the modified areas is indicated in red.]

 Comments 5: [The abstract should be revised to clearly state the objective of the review and summarize key comparative insights. It currently reads as a list rather than a focused summary and lacks mention of key findings or future directions]

 Response 5: [Thank you very much for the professional and valuable feedback from the reviewer.The modifications have been made, and the font color in the modified areas is indicated in red]

 Comments 6: [The author can enrich the literature review on digital twin applications in damage detection by citing the following article: Genetic Algorithm-Based Model Updating in a Real-Time Digital Twin for Steel Bridge Monitoring]

 Response 6: [Thank you very much for the professional and valuable feedback from the reviewer.The reference has been cited, and the font color for the modifications is indicated in red.]

Reviewer 2 Report

Comments and Suggestions for Authors

This research is expected to have a great influence on various infrastructure environments in the train industry and tracks. In addition, this paper is well written and an interesting subject.

The following modifications are required bellows;

  • Are the MDPI reference forms compliant?
  • Are the cited references mostly recent publications (within the last 5 years) and relevant?
  • Are the conclusions consistent with the evidence and arguments presented?
  • About Section 2, It is not enough related work, so it is required qualitative research.
  • Show the numeric result for testing from 3.1 to 3.4
  • We understood the description about the Railway Non destructive Testing Platform and Equipment, however, it must have a quantitative experimental results about Section 4.

Author Response

Comments 1: [Are the MDPI reference forms compliant?]

Response 1: [Thank you very much for the valuable feedback from the reviewer.The table was added and written according to the format requirements provided by MDPI. Meets the format requirements of MDPI.]

Comments 2: [Are the cited references mostly recent publications (within the last 5 years) and relevant?]

Response 2: [Thank you very much for the valuable feedback from the reviewer.I have added literature from the past 5 years or so, and the font color for the modifications is indicated in red.]

Comments 3: [Are the conclusions consistent with the evidence and arguments presented?About Section 2, It is not enough related work, so it is required qualitative research.]

Response 3: [Thank you very much for the valuable feedback from the reviewer.The modifications have been made, and the font color in the modified areas is indicated in red.]

Comments 4: [Show the numeric result for testing from 3.1 to 3.4.]

Response 4: [Thank you very much for the valuable feedback from the reviewer.The modifications have been made, and the font color in the modified areas is indicated in red.]

Comments 5: [We understood the description about the Railway Non destructive Testing Platform and Equipment, however, it must have a quantitative experimental results about Section 4.]

Response 5: [Thank you very much for the valuable feedback from the reviewer.The modifications have been made, and the font color in the modified areas is indicated in red.]

Reviewer 3 Report

Comments and Suggestions for Authors

The topic chosen by the authors is of great importance, given technological advances and the need for faster transportation in large urban centers. With society's ever-increasing demand for the number of trips throughout the day and the demand for high speed, the materials that make up rapid transit systems can be pushed to the limit of fatigue.

Therefore, monitoring techniques need to be constantly improved to detect these extreme situations before they cause accidents. The review manuscript entitled “Review on detection technologies of rail damage high-speed 2 trains” The authors reported a series of non-destructive tests as a way of monitoring the physical wear conditions of materials present in high-speed train lines, which is indeed interesting, but without wishing to belittle the authors' work, in order for the work to have robust results, a number of studies with infinite wear conditions could be included to associate with the monitoring, which is reflected in the small number of citations throughout the text. Thus, to be a good review paper, the authors need to be careful about what publications are most relevant to the subject, which was not done, as can be seen in the publication https://doi.org/10.3390/app13084790 review article entitled " A Review of Fault Diagnosis Methods for Key Systems of the High-Speed Train]” published in 2023 by the same journal, Applied Sciences, which is not included in the citations. Note that the authors took care to cite a significant number of works, with almost 100 references.

Even with few works analyzed, the authors made an interesting comparison with robust results from relevant publications, as can be seen in Table 1. I suggest that the authors expand the work to not only detect a situation of fatigue or collapse, but also to add other factors associated with the quality of the materials used, with various techniques applied in mechanical engineering for manufacturing materials and material performance.

 I recommend reformulating the conclusion to have a clearer connection with the entire text presented, the conclusion of the work based on the data presented, in sentences 452 to 467. Even though AI tools are a reality today, the authors need to present concrete data on how this tool can be used.

Author Response

Comments 1:[Thus, to be a good review paper, the authors need to be careful about what publications are most relevant to the subject, which was not done, as can be seen in the publication https://doi.org/10.3390/app13084790 review article entitled " A Review of Fault Diagnosis Methods for Key Systems of the High-Speed Train]” published in 2023 by the same journal, Applied Sciences, which is not included in the citations. Note that the authors took care to cite a significant number of works, with almost 100 references.]

Response 1: [Thank you very much for the valuable feedback from the reviewer.Modifications have been made and the reference has been cited. The font color for the modifications is indicated in red.]

Comments 2: [Even with few works analyzed, the authors made an interesting comparison with robust results from relevant publications, as can be seen in Table 1. I suggest that the authors expand the work to not only detect a situation of fatigue or collapse, but also to add other factors associated with the quality of the materials used, with various techniques applied in mechanical engineering for manufacturing materials and material performance.]

Response 2:[Thank you very much for the valuable feedback from the reviewer.Table 1 has been modified and relevant data has been added. The font color for the modified areas is indicated in red.]

Comments 3:[ I recommend reformulating the conclusion to have a clearer connection with the entire text presented, the conclusion of the work based on the data presented, in sentences 452 to 467. Even though AI tools are a reality today, the authors need to present concrete data on how this tool can be used.] 

Response 3: [Thank you very much for the valuable feedback from the reviewer.The modifications have been made, and the font color in the modified areas is indicated in red.]

Round 2

Reviewer 1 Report

Comments and Suggestions for Authors

The revised submitted manuscript is suitable for publication. No further comments.

Reviewer 3 Report

Comments and Suggestions for Authors

The authors successfully carried out all the requested recommendations, as well as received a more robust and current version for a review article on the proposed topic, therefore consider that the manuscript in its current version is eligible for publication.